# Prediction of Urban Sprawl by Integrating Socioeconomic Factors in the Batticaloa Municipal Council, Sri Lanka

Mathanraj Seevarethnam [1,2,*], Noradila Rusli [3] and Gabriel Hoh Teck Ling [1]

1. Department of Urban and Regional Planning, Faculty of Built Environment and Surveying, Universiti Teknologi Malaysia, Johor Bahru 81310, Johor, Malaysia
2. Department of Geography, Faculty of Arts and Culture, Eastern University, Vantharumoolai 30350, Sri Lanka
3. Centre for Innovative Planning and Development (CIPD), Department of Urban and Regional Planning, Faculty of Built Environment and Surveying, Universiti Teknologi Malaysia, Johor Bahru 81310, Johor, Malaysia
* Correspondence: mathanrajs@esn.ac.lk

**Abstract:** Due to extensive population growth, urbanization increases urban development and sprawl in the world's cities. Urban sprawl is a socioeconomic phenomenon that has not extensively incorporated socioeconomic factors in the prediction of most of the urban sprawl models. This study aimed to predict the urban sprawl pattern in 2030 by integrating socioeconomic and biophysical factors. NDBI, Cramer's V, logistic regression, and CA-Markov analyses were used to classify and predict built-up patterns. The built-up area is the dominant land use, which had a gradual growth from 1990 to 2020. A total of 20 socioeconomic and biophysical factors were identified as potentials in the municipality, affecting the urban sprawl. Policy regulation was the most attractive driver with a positive association, and land value had a high inverse association. Three prediction scenarios for urban sprawl were achieved for 2030. Higher sprawling growth is expected in scenario 3, compared with scenarios 1 and 2. Scenario 3 was simulated with biophysical and socioeconomic factors. This study aids in addressing urban sprawl at different spatial and temporal scales and helps urban planners and decision makers enhance the development strategies in the municipality. Predicted maps with different scenarios can support evaluating future sprawling growth and be used to develop sustainable planning for the city.

**Keywords:** urban sprawl; urban modeling; land use; urbanization; socioeconomic factors

## 1. Introduction

Urbanization and rampant population growth in urban areas are increasing urban development in the world's cities today. The highest proportion of urbanization occurred in the first decade of the 21st century [1]. The urban population is more than half of the world's population today, which creates remarkable challenges for urban planners and governments. Most cities in developing countries have experienced one challenge, which is urban sprawl [2–5]. Urban sprawl is a controversial topic among researchers who still do not agree on an acceptable definition. Urban sprawl is a term for many circumstances because its characteristics are not parallel in cities around the world [2,6,7]. Thus, it is called a multidimensional phenomenon and an ambiguous concept [8–10].

Urban sprawl is a specific urban environment that should be linked to the necessity for sustainable development goals [4]. Sprawl is a socioeconomic phenomenon that should be understood in various socioeconomic contexts to move towards sustainability. For example, individual housing development increases the land value in cities, which has a direct effect on the growth of urban sprawl. Thus, the land value is one of the influencing socioeconomic phenomena in many cities around the world that should be considered for sustainable land utilization. A limited number of biophysical factors are in the world such as slope, elevation, and climate conditions that do not always take place for urban sprawl

in all cities. However, many socioeconomic factors cause urban sprawl development, but it is hard to realize which factors have the most impact. These factors vary between cities, regions, and countries in the world [11] and are influenced by land values, demographic patterns, topographic landscape, societal factors, and means of transport [12]. Limited studies were conducted on the socioeconomic factors of urban sprawl in developed [13–15] and developing [4,16–18] countries. More precisely, a comprehensive study covering many socioeconomic factors was not identified in any urban areas that should be focused on to fill the gap. Thus, this study covered several socioeconomic factors, such as housing preference, income inequality, demographic dynamics, land value, education, employment, regional accessibility, living standard, and policy regulation, which were significantly identified in different cities in the world [8,19–22].

Furthermore, the urban model has a long history from the growth of the first computing model in the urban domain [23,24]. The urban model describes, builds, and uses spatial models for specific reasons in traditional spatial design. In the last 50 years, different types of urban models have been developed globally [25], which are categorized from several perspectives. Later, it was extended to spatial analysis to develop location theories in social and human geographies. Finally, these models are practiced to test theories related to spatial location and find the connection between land use and its activities [24]. Different authors categorize urban models into different groups [24–26]. For example, Bhatta classified the model into three groups, which are (1) the agent-based model, cellular automata model, and micro-simulation model, (2) land use transportation model, and (3) urban dynamics models. Of these, group one (1) models have two diverse-type models, such as the agent-based model and cellular automata model, comprising the same group. Simultaneously, the neural network model and fractal-based model were not considered in these groups. In contrast, Pooyandeh et al. [25] categorized the spatial and temporal urban models into two groups such as the complexity and temporal GIS models. These two models were also divided into many different subgroups. Additionally, these models have been overlooked in the theoretical models [24]. However, Márquez et al. [26] divided the prediction models for urban sprawl and land use into three groups, which are the GIS-based models, machine learning models, and hybrid models.

GIS-based models are usually used for the prediction of land use/land cover patterns. For example, STSM [27], SLEUTH [28], LTM [29], and CLUE [30] models are widely used GIS-based models in urban sprawl, urban growth, and land use studies. These models encode the data to develop spatial layers of predictor variables. These variables can be generated from a series of land use maps that use spatial rules to relate to these variables for land use transformation. Then, a time series map can be generated according to the amount of transition scale with future land use [26]. Then, machine learning models are the vital component in artificial intelligence that mainly uses computer algorithms for the simulation and prediction of land use/land cover. Markov chain [31], cellular automata [32], logistic regression [33], multi-agent model [34], ANN [35], and SVM [5] are some models that often inspire academics to apply for urban studies. Furthermore, they can function without human assistance and are also familiar with mathematical and statistical approaches. Machine learning models are broadly applied in urban studies due to the efficiency and precision of this model. In addition, it has an important feature that has the ability to use driving factors in land use/land cover predictions [36]. Furthermore, hybrid models are theories based on systems with multi-scale features. For example, CLUE-S-MC [37], CA-MC [38], ANN-CA [39], MLP-MC [40], and LR-MC-CA [41] are some of the models used in previous studies for the prediction of land use or urban expansion. Different land use change scenarios are analyzed at different spatial scales using these models. These models fit well with other models and are used to understand the relationship between land use change and the driving factors. It uses a top-down approach with a bottom-up method to integrate the land use patterns and driving factors [26]. Additionally, socioeconomic driving factors were integrated with the land use prediction in past studies. Moreover,

the different datasets' inconsistency is a major challenge to applying these factors in the models [37].

In this case, past studies were not widely applied to much of the model of urban sprawl in developed and developing countries [24]. The applied models were not focused on many socioeconomic factors, which are the essential phenomena for urban sprawl in the city. In the Sri Lankan context, urban sprawl studies are limited to spatiotemporal pattern analysis in larger cities such as Kandy [42] and Colombo [43] in Sri Lanka. Nevertheless, predictions of urban sprawl are missing at the national level, including large cities such as Colombo or Kandy. Due to this lack of prediction on urban sprawl, it will be more possible to increase sprawl in cities in the future. Increased population growth can increase the need for residential buildings, which can further increase land values in urban and suburban areas. This will lead to the expansion of the suburbs and the growth of urban sprawl in the suburbs. As socioeconomic factors are strongly influenced by urban sprawl in cities, this study is more significant because it incorporates nine socioeconomic factors identified from various cities around the world that are included for predicting urban sprawl in 2030. Thus, this study can be a direction for the urban sprawl studies in the cities of Sri Lanka and developing countries in the world. Therefore, this study aimed to predict the urban sprawl pattern in 2030 by integrating socioeconomic and biophysical factors.

## 2. Materials and Methods

### 2.1. Study Area

The Batticaloa Municipal Council is one of the local authorities in the Batticaloa district of Sri Lanka. The municipality is located on the eastern coast of Sri Lanka with an average elevation of 8523 meter mean sea level (see Figure 1). The total population is 94,131, and the density was 1372 people per square kilometer in 2020 [44]. The total extent of Batticaloa municipality is 4195.18 hectares used for various uses. Residential land use is the most dominant in this area, with around 1170.24 hectares, followed by agricultural land, with about 935.6 hectares. Commercial land use is one of the low extents, approximately 23.5 hectares. In addition, natural area (82.5 hectares), scrublands (185.71 hectares), and water bodies (58 hectares) have a certain percentage in this area.

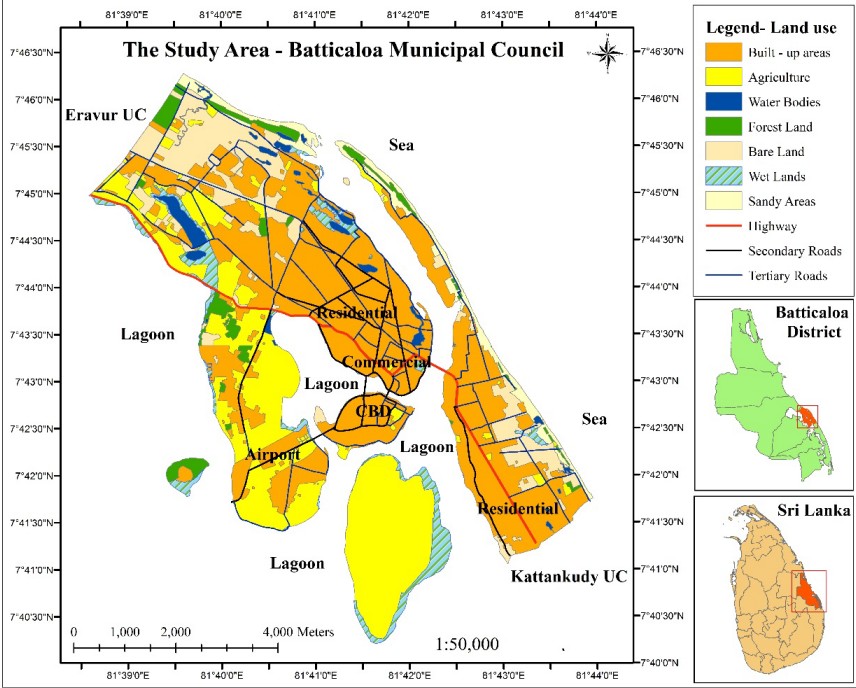

**Figure 1.** The study area: Batticaloa Municipal Council. Source: modified from Batticaloa MC, 2020.

The Batticaloa area was hit severely by civil war for three decades until 2009. Its inhabitants began to enjoy a peaceful life after the end of the civil war in Sri Lanka. Several development projects have started due to the peaceable environment in this area. It encourages investors, donors, and non-governmental organizations integrated with the government to rebuild the city. The prompt reconstructions create many socioeconomic problems due to the built-up growth that creates sprawling development in the city. Therefore, this area has to consider studying urban sprawl in order to develop a sustainable city in Sri Lanka.

### 2.2. Data Source

Landsat images for the years 1990, 2000, 2010, and 2020 with all bands were downloaded from the Earth Explorer, United States Geological Survey (see Table 1). These downloaded images were projected to the Kandawala local coordinate system. The Batticaloa municipality area was digitized based on the existing municipality boundary, Batticaloa, and the Landsat images were clipped using this boundary shapefile for the analysis. Elevation and slope were obtained from Google Earth Pro. Water bodies; highways, secondary roads, and tertiary roads; railways and airline runways; CBD; commercial, educational, administration, religious, and services buildings; population by GN division; and land value by GN division were digitized in Google Earth Pro based on the existing maps in Batticaloa municipality. Socioeconomic factors such as land value, demographic dynamics, policy regulations, regional accessibility, housing preferences, income inequality, education, living standards, and employment were collected with geographic coordinates through a questionnaire survey in the Batticaloa municipality to develop the maps.

**Table 1.** Details of satellite images.

| Satellite Name (Year) | Satellite Image ID | Acquisition Date |
| --- | --- | --- |
| Landsat 5 TM (1990) | LT05_L1TP_140055_19900912_20200915_02_T1 | 12 September 1990 |
| Landsat 7 ETM+ (2000) | LE07_L1TP_140055_20000928_20200917_02_T1 | 28 September 2000 |
| Landsat 7 ETM+ (2010) | LE07_L1TP_140055_20100924_20200910_02_T1 | 24 September 2010 |
| Landsat 8 OLI (2020) | LC08_L1TP_140055_20200927_20201005_02_T1 | 27 September 2020 |

Source: Earth Explorer (2021) [45].

### 2.3. Data Analysis

Landsat images were employed with the following methods to analyze and interpret the results and findings.

#### 2.3.1. Normalized Difference Built-Up Index (NDBI)

NDBI is used to extract built-up features from a surface image with indices ranging from −1 to 1. The built-up land shows a positive value, and other land use shows negative values [46]. This method is taken to the benefit of the individual spectral responses of built-up lands and other land covers. NDBI is calculated using Equation (1) [47]:

$$NDBI = (SWIR - NIR)/(SWIR + NIR) \tag{1}$$

where NDBI indicates the normalized difference built-up index, SWIR is the shortwave infrared band, and NIR is the near-infrared band.

The built pattern for the years 1990, 2000, 2010 and 2020 was extracted to understand the growth of the urban sprawl in Batticaloa municipality. The built-up area increased gradually since the beginning due to the rapid growth in construction as the municipality became an urban area. This building growth was experienced in the last three decades. Therefore, earlier (1990) and later (2020) built-up images were used in the change analysis in Land Change Modeler, IDRISI 17.0, which has a change analysis and prediction tools. A map of built-up changes was generated using this tool to understand the growth of urban

sprawl for the selected years. Then, this map was used in the logistic regression model to develop a transition potential map.

### 2.3.2. Socioeconomic Data Transformation

A total of nine socioeconomic factors were identified from previous studies related to urban sprawl, which are housing preference, income inequality, demographic dynamics, land value, education, employment, regional accessibility, living standard, and policy regulation. A questionnaire survey was developed to obtain the status of these factors from 400 households' respondents in Batticaloa municipality. These 400 questionnaires were collected with the geographical coordinates of the location where the respondent filled in the form to understand the level of influence of each socioeconomic factor (see Figure 2). Batticaloa municipality has approximately 123,750 pixels with 30 m resolution in the datasets. Of these, 400 random pixels (random points) were selected from the field survey to complete the questionnaires. These 400 random points were then marked in Google Earth Pro, which was imported into ArcGIS to obtain the spatial map of socioeconomic factors (see Figure 3). According to the survey numbering, each questionnaire was joined with these 400 points using the Join Attribute tool in ArcGIS. Then, each factor was mapped with the details from the questionnaire on how each socioeconomic factor can be transformed for each point on the map. For example, if a factor showed (0), the factor does not influence the urban sprawl in that area; if a factor showed (1), the factor has a 'low influence' in the specific area; if a factor showed (2), the factor has a 'moderate influence' in the particular area; if a factor showed (3), the factor 'strongly influenced' the urban sprawl in a specific area. Table 2 presents an example of the distribution of the questionnaire data.

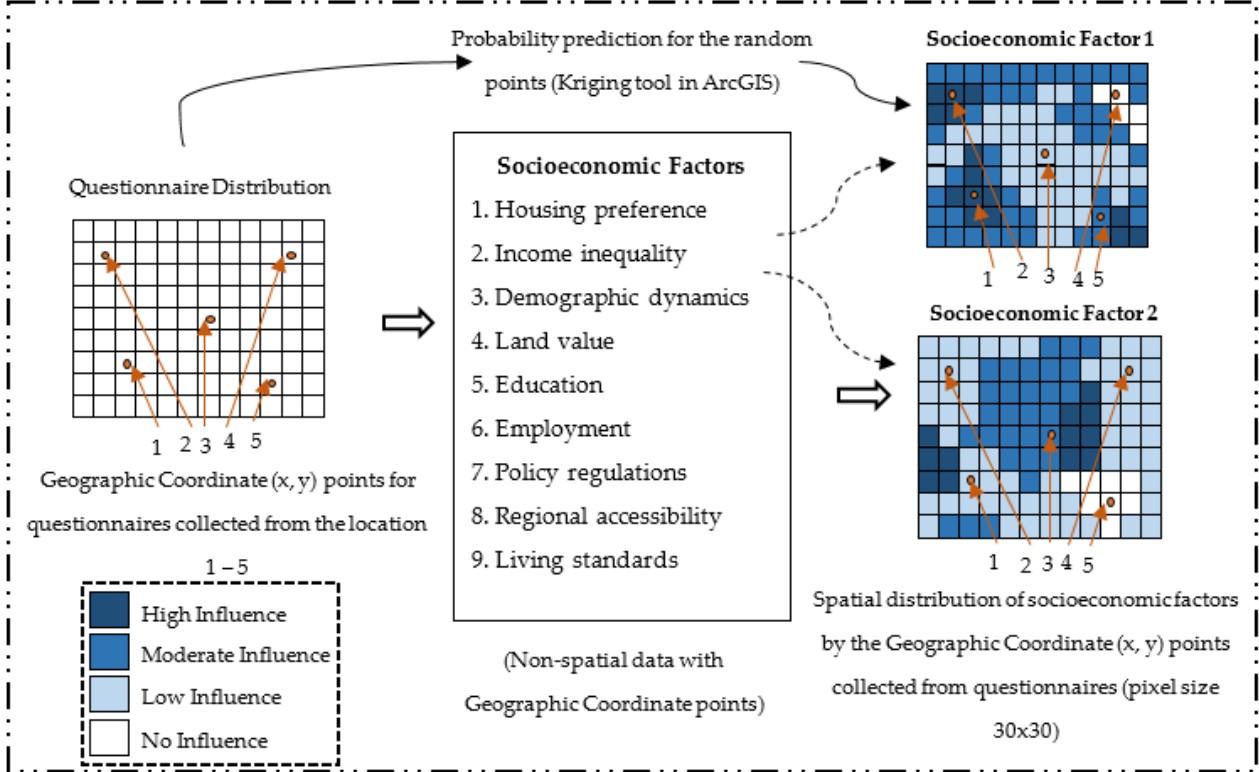

**Figure 2.** The attribute data transformation to spatial maps.

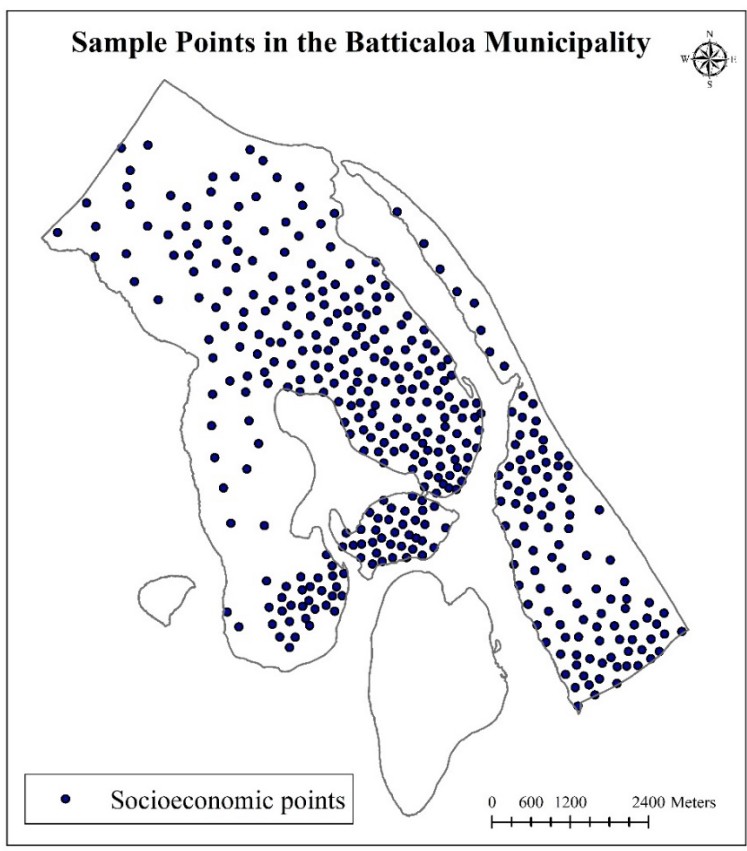

**Figure 3.** Sample points of questionnaires for socioeconomic factors.

**Table 2.** Example of the distribution of socioeconomic factors in attribute data.

| Point No. (Questionnaire No.) | Housing Preference | Income Inequality | Demographic Dynamics | Land Value | Education | Employment | Policy Regulations | Regional Accessibility | Living Standards |
|:---:|:---:|:---:|:---:|:---:|:---:|:---:|:---:|:---:|:---:|
| 1 | 3 | 1 | 0 | 1 | 1 | 1 | 3 | 1 | 1 |
| 2 | 1 | 2 | 1 | 2 | 2 | 1 | 2 | 2 | 2 |
| 3 | 0 | 1 | 1 | 3 | 2 | 3 | 1 | 2 | 3 |
| 4 | 1 | 2 | 3 | 3 | 3 | 1 | 1 | 2 | 2 |
| 5 | 2 | 1 | 1 | 2 | 3 | 1 | 2 | 3 | 2 |

These 400 samples were mapped using the kriging interpolation tool in ArcGIS. Kriging produces the probability features for points in a neighborhood around each output raster cell (30 × 30 pixel size). Each point has a smooth curved surface that shows the highest value of this surface at the location point. It drops off as the distance from the point increases and becomes zero at the search radius distance from the point. This method overcomes many shortcomings of traditional interpolation methods in ArcGIS. Moreover, the assumptions of this theory are less random than other methods, which gives a reliable probability. As an optimal interpolator, the model estimates are objective and the minimum variance is known. The variogram and data settings define the weight of the kriging method. While the variances of the estimate are determined and mapped as estimates, a specific distribution can be assumed. Thus, kriging differs from other interpolators [48].

### 2.3.3. Cramer's V Test

Interpolated maps of socioeconomic factors and other biophysical factors were tested with Cramer's V to identify potential drivers of urban sprawl. When the Cramer's V value is close to 0, the association between the variables is lower. Simultaneously, when the value is close to 1, the association between the variables is strong. However, a V value higher than 0.15 is useful and greater than 0.4 is considered good [49,50]. Thus, the prediction model only included the explanatory factors of a value higher than 0.15. Approximately 26 variables were tested with Cramer's V in the Land Change Modeler of IDRISI 17.0. A total of 20 factors were identified as having potential with a value of V higher than 0.15. Therefore, these variables were included in the logistic regression model to develop a probability surface.

### 2.3.4. Sub Model: Logistic Regression

As a sub-model, logistic regression was used to identify potential transition areas with Land Change Modeler, which is a combined module in IDRISI 17.0. This module is considered to be a good platform with various integrated models such as logistic regression, cellular automata, and Markov chain [49]. This model runs a set of transitions with the potential factors identified from the Cramer's V test. The results of this transition were expressed in the coefficient value of the factors with a transition potential map. This coefficient value mentions the effects of each independent variable (socioeconomic factors) on the dependent variable (built-up). It varies between −1.0 (strong negative association) to +1.0 (strong positive association). A coefficient value of less than 0.4 indicates that there is no redundancy between the variables [49,51].

### 2.3.5. CA-Markov Model

Built-up maps and a transition potential map were used to predict the urban sprawl pattern in 2030. The Markov model is capable to monitor, simulate, and predict the land use pattern. A Markovian transition estimator was used to compute the transition probability matrix for a built-up change from 1990 to 2020. The prediction is calculated as follows (see Equations (2) and (3)):

$$S(t, t+1) = Pij \times S(t) \tag{2}$$

where S (t) is the system status at the time of t, S (t + 1) is the system status at the time of t + 1, and Pij is a transition probability matrix of state calculated as follows:

$$|| \, || = Pij = \begin{Vmatrix} P_{1,1} & P_{1,2} & \dots & P_{1,N} \\ P_{2,1} & P_{2,2} & \dots & P_{2,N} \\ \dots & \dots & \dots & \dots \\ P_{N,1} & P_{N,2} & \dots & P_{N,N} \end{Vmatrix} \quad 0 \leq Pij \leq 1 \tag{3}$$

where $P$ is the transition probability, $P_{ij}$ represents the conversion probability from the existing state $i$ to another state $j$ in the next time, and $P_N$ is the state's probability for any time. A probability close to (0) specifies a low transition, and probabilities close to (1) designate a high transition [52–54].

The transition probability for the period of 1990–2020 was calculated to predict the urban sprawl in 2030. Cellular automata is a dynamic model that is familiar with analyzing a built-up pattern with the Markov model. This integrated CA-Markov model permits us to simulate a two-way transition and predict the transition between any number of classes [55]. The CA model is expressed as follows (see Equation (4)):

$$S(t, t+1) = f(S(t), N) \tag{4}$$

where S (t + 1) is the system status at the time of (t, t + 1) operated by the state probability at all times (N).

2.3.6. Model Validation

The model was validated with validation from IDRISI for the simulated and existing maps for 2020. Three parameters are in the IDRISI validation tool. Kno is the overall kappa accuracy, Kstandard is the quantity, and Klocation is the kappa for location, which are greatly recommended to evaluate the simulated maps. A kappa value equal to 1 is satisfactory, and a value equal to 0 is unsatisfactory [55]. Based on the performance of kappa results, the prediction of future patterns can be displayed in the urban sprawl. The model accuracy of nearly 0.80 is typically considered adequate and an excellent agreement of the map data [49]. Thus, the predicted map for 2030 can be considered to be accurate. Furthermore, the relative operating characteristic (ROC) in logistic regression was used to compare the suitability map with the Boolean map of "reality" and determine the best cutoff value. It contains an outstanding statistic to compute the goodness of fit of the logistic regression model. The ROC value ranges between zero and one in the model. One defines the success of the model, and 0.5 states a random fit of the model [49]. In addition, the predictive strength of the logistic regression model was assessed with pseudo R2 statistics. Pseudo R2 close to 1 specifies a perfect fit, and pseudo R2 close to 0 establishes no relationship. Nevertheless, Pseudo R2 greater than 0.2 is considered a relatively good fit [50]. Therefore, ROC and Pseudo R2 were calculated to identify the fitness of the model.

## 3. Results and Discussion

### 3.1. Built-Up Pattern

As the dominant land use in Batticaloa municipality, the built-up area has experienced rapid growth in recent decades. The built pattern is mainly considered the parameter for identifying the urban sprawl in a city. Hence, it was achieved for the years 1990, 2000, 2010, and 2020 (see Figure 4). The built-up area was 1030.23 hectares in 1990, 1237.86 hectares in 2000, 1367.10 hectares in 2010, and 1774.44 hectares in 2020 (see Table 3). The accuracy level was 93% in 1990, 94% in 2000, 93% in 2010, and 94% in 2020. The built-up pattern showed a continuous gradual increase in the built-up area during this period. However, the municipality has faced sprawling growth since 1990 that was establish based on these built-up maps. As a central area in the eastern province, Batticaloa experienced the effects of civil war from the 1980s to 2000s. This context was one of the factors that caused human migration to and from other districts to access security, food, health, education, and other services. For instance, the International Organization for Migration reported in 2007 that the civil war caused more than 150,000 people to flee to eastern Batticaloa, especially to Batticaloa municipality. A total of 750 shelters were supplied to the residents of Batticaloa municipality and the Chenkalady area at the same time as construction on 700 more shelters began in 2007 [56]. Moreover, the increase in residential and commercial buildings caused the growth of urban built-up areas during these periods. Therefore, the residents contributed significantly to this rapid urban growth in the municipality.

**Table 3.** The extent of built-up area in Batticaloa municipality (ha).

| Class Name | 1990 | % | 2000 | % | 2010 | % | 2020 | % |
|---|---|---|---|---|---|---|---|---|
| Built-up | 1030.23 | 24.56 | 1237.86 | 29.51 | 1367.10 | 32.59 | 1774.44 | 42.30 |
| Non-built-up | 3164.95 | 75.44 | 2957.32 | 70.49 | 2828.08 | 67.41 | 2420.74 | 57.70 |
| **Total** | **4195.18** | | **4195.18** | | **4195.18** | | **4195.18** | |

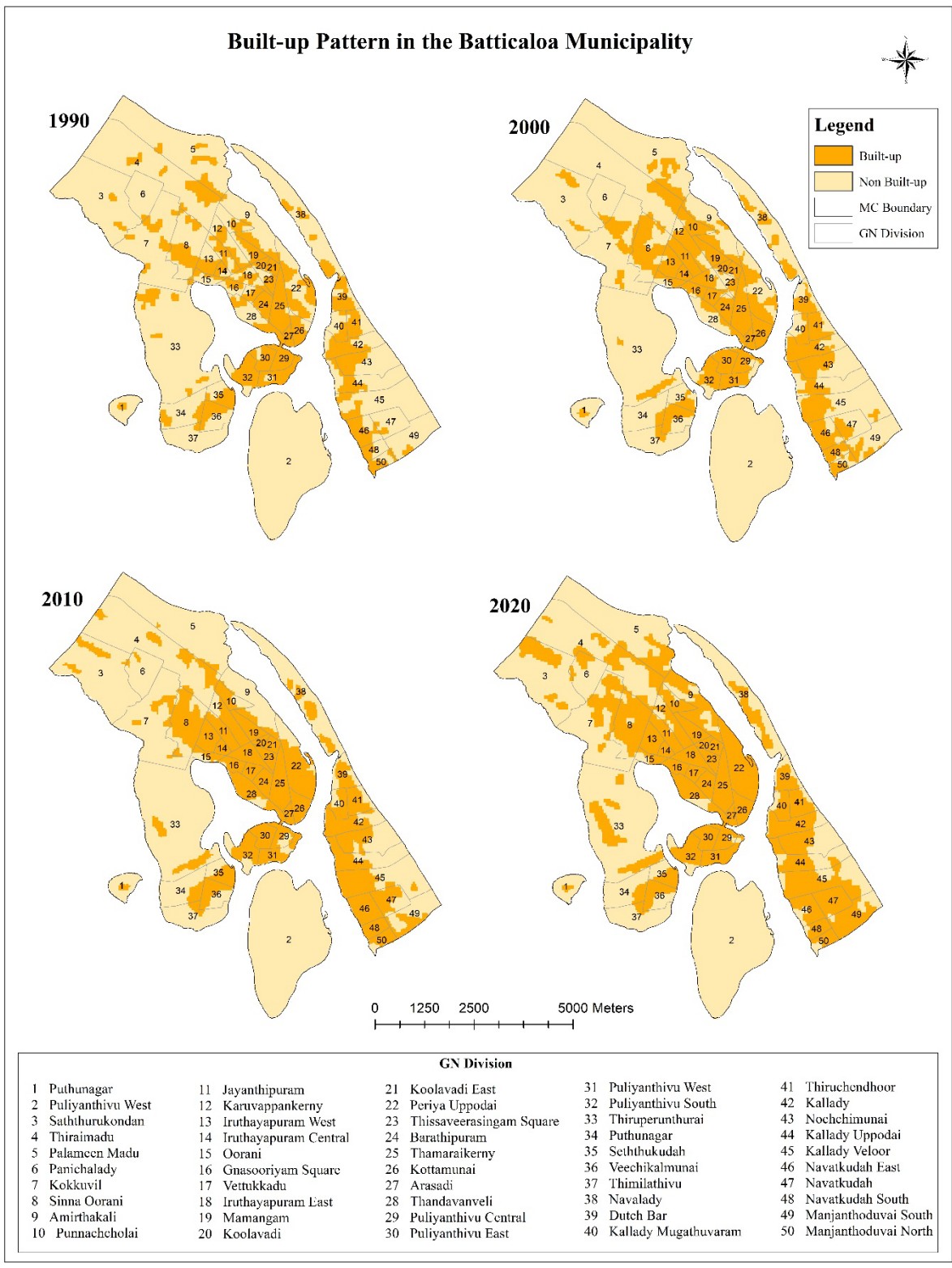

**Figure 4.** Built-up pattern for the years 1990, 2000, 2010, and 2020.

The gains and losses of built-up areas between 1990 to 2020 were identified based on the built-up maps (see Figure 5). The built-up area gained 906 hectares and lost 162 hectares in these periods. Replacement of residents due to the tsunami and flood in the municipality was a notable factor for the losses of the built-up in coastal areas.

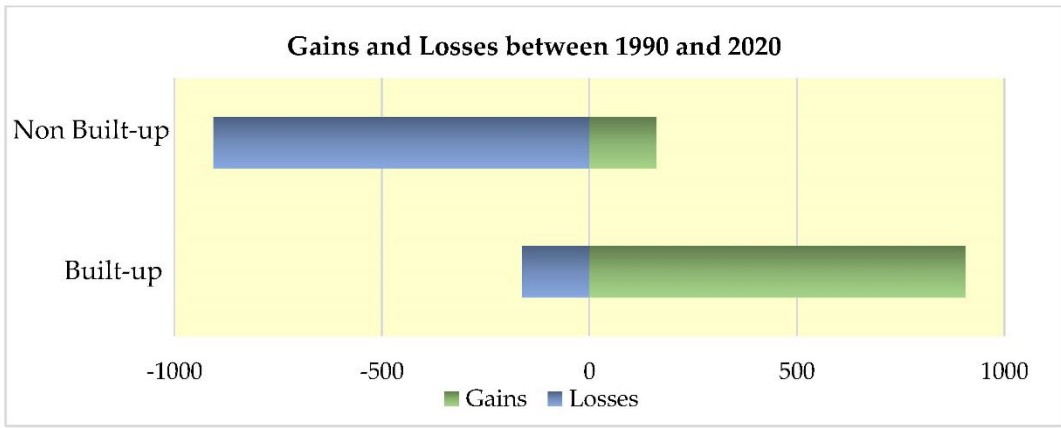

**Figure 5.** Gains and losses of the built-up area between 1990 and 2020 (in hectares).

*3.2. Selection of Drivers' Variables*

A total of 26 biophysical and socioeconomic factors related to urban sprawl were identified from the literature of developed and developing countries (see Figure 6). First, it was determined whether these variables in this study were appropriate to use in the Batticaloa area. Batticaloa is a coastal region with inland water bodies; hence, physical factors such as elevation, slope, and water bodies are important factors in built-up development in the municipality. Land value and population growth have a significant relationship with urban sprawl in Batticaloa as it is a growing city. Policy regulations, people's housing preferences, income inequality, and living standards all have a connection to urban sprawl. Education and employment are important factors for the sprawling growth of the city. Regional accessibility, transport accessibility, and service buildings, including commercial, CBD, administration, and religious buildings, are primarily related to residential growth. Therefore, these factors were focused on in this study. Then, these factors were tested with Cramer's V, available in Land Change Modeler, IDRISI 17.0. The potential factors of urban sprawl were identified with the built-up patterns in the Batticaloa municipality. These factors were used to develop the transition potential for urban sprawl prediction.

Factors with a Cramer's V value greater than 0.15 were considered for the prediction of urban sprawl in 2030. In this case, 20 variables (factors) met this requirement that was added to the logistic regression model. Six socioeconomic variables, such as housing preferences, income inequality, education, living standard, employment, and distance to the air runway, did not meet Cramer's V requirements (see Table 4). These variables were not taken into account for predicting urban sprawl patterns. Furthermore, several factors were considered in this study, which many authors ignored due to data availability. For example, Mustafa and Teller [57] indicated that their study was limited to several urban sprawl factors in the model. They did not include accessibility to roads, trains, and jobs, which is a hidden factor that should obviously affect urban sprawl over time. Simultaneously, Saqui [58] and Grigorescu et al. [30] mentioned that several explanatory variables identified in previous studies were not included in their research studies due to a lack of data. Therefore, the factors used in this study can support the urban sprawl prediction in all aspects.

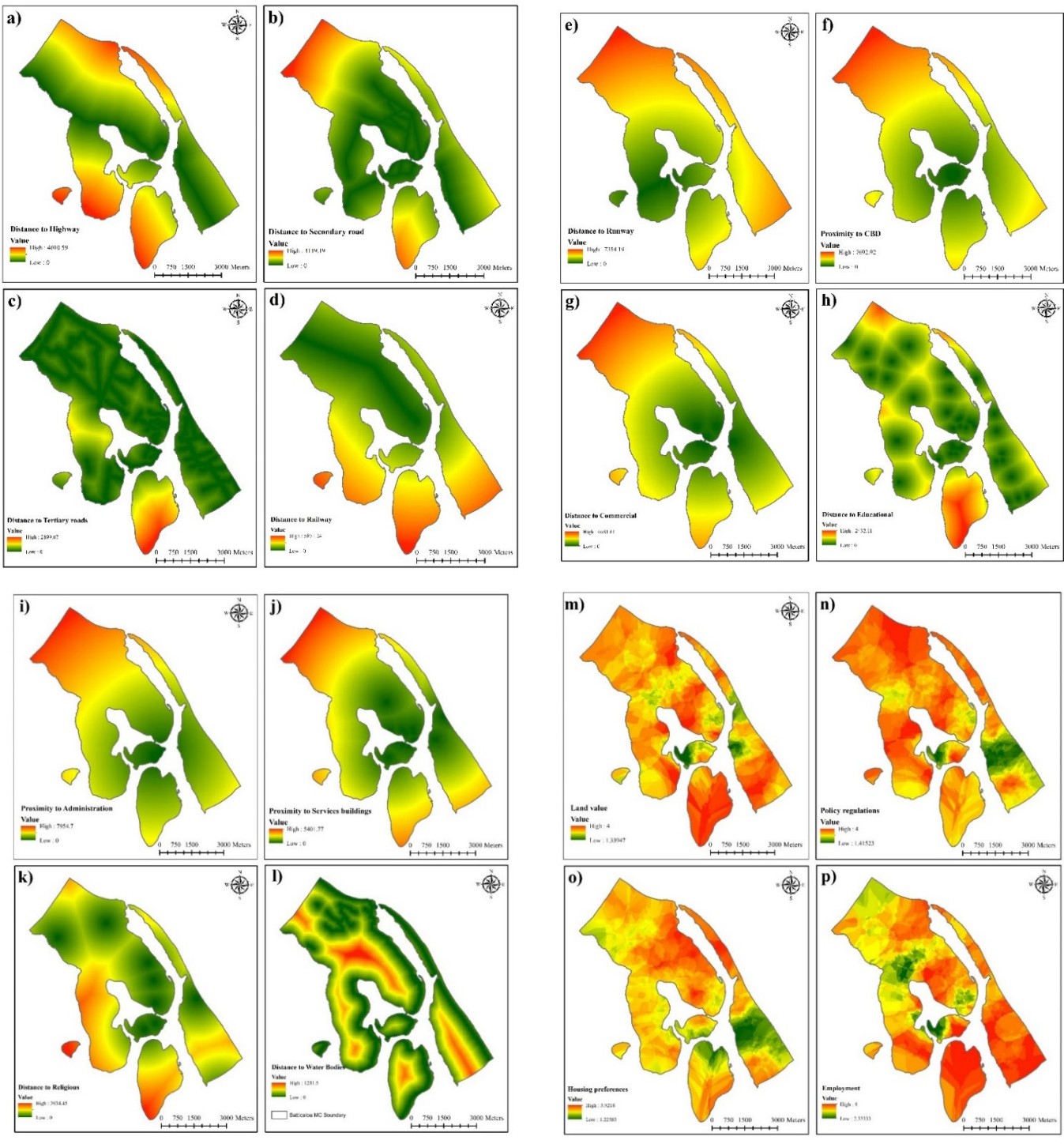

**Figure 6.** *Cont.*

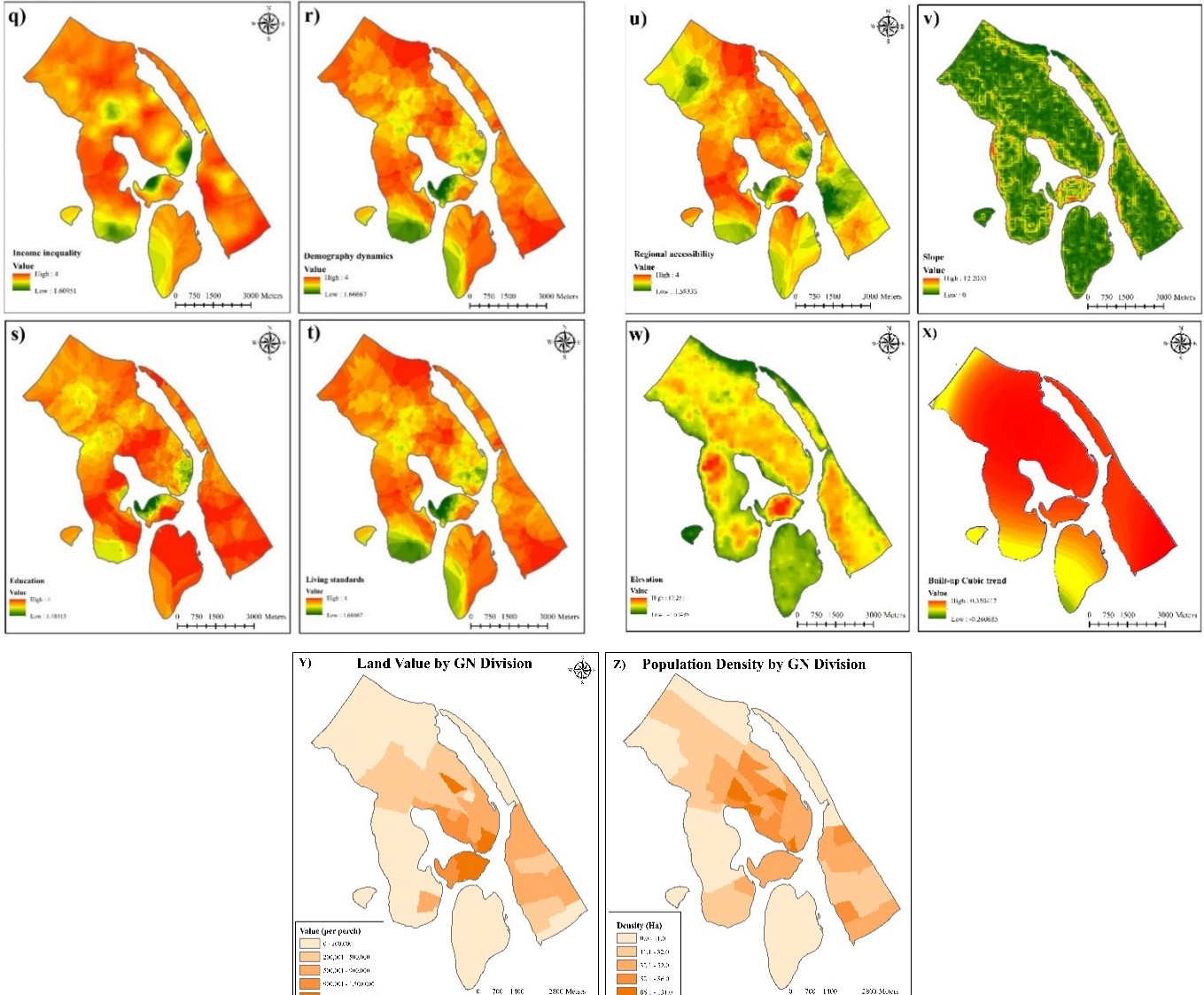

**Figure 6.** Socioeconomic and biophysical factors related to urban sprawl (**a–z**).

**Table 4.** Cramer's V and coefficient values for biophysical and socioeconomic factors.

| | Variables | Cramer's | Coefficient | | | Variables | Cramer's | Coefficient |
|---|---|---|---|---|---|---|---|---|
| **(a)** | **Biophysical Factors** | | | | 14 | Employment | 0.0756 | |
| 1 | Elevation | 0.4260 | 0.4615 | | **(b1)** | **Transport Accessibility** | | |
| 2 | Slope | 0.1531 | −0.4546 | | 15 | Distance to highway | 0.5192 | 0.00000443 |
| 3 | Distance to water bodies | 0.1903 | 0.0009572 | | 16 | Distance to secondary road | 0.4877 | 0.000083 |
| **(b)** | **Socioeconomic Factors** | | | | 17 | Distance to tertiary road | 0.3398 | 0.0027 |
| 4 | Land value | 0.1980 | −0.7519 | | 18 | Distance to railway | 0.2553 | 0.00008454 |
| 5 | Demographic dynamics | 0.1956 | −0.5519 | | 19 | Distance to air runway | 0.1375 | |
| 6 | Policy regulations | 0.1603 | 0.9036 | | **(b2)** | **Services Buildings** | | |
| 7 | Regional accessibility | 0.1537 | 0.7254 | | 20 | Proximity to CBD | 0.2322 | −0.0016 |
| 8 | Population by GN division | 0.4620 | −0.5766 | | 21 | Distance to commercial | 0.2839 | 0.0000318 |
| 9 | Land value by GN division | 0.5113 | 0.000491 | | 22 | Distance to educational | 0.3478 | −0.0012 |
| 10 | Housing preferences | 0.1478 | | | 23 | Proximity to administration | 0.2240 | 0.0009856 |
| 11 | Income inequality | 0.1376 | | | 24 | Distance to Rrligious | 0.3624 | −0.0015 |
| 12 | Education | 0.0783 | | | 25 | Proximity to services buildings | 0.2844 | 0.0012 |
| 13 | Living standards | 0.0956 | | | 26 | Built-up cubic trend | 0.5462 | 3.6418 |

(Note: Cramer's V value greater than 0.15 was considered for the urban sprawl prediction.).

The Cramer's V and coefficient values for the factors from the logistic regression analysis are presented in Table 4. The results indicated that the spatial associations between the explanatory variables and urban sprawl differed in relation to the biophysical and socioeconomic factors. The coefficient value of the logistic regression denoted the individual influence of each factor on the transition variable with direct or inverse associations [49]. The coefficients specified which biophysical and socioeconomic variables had the most vital contribution to describe the spatial pattern of built-up areas. The positive coefficient value for the variables indicated the direct association with urban sprawl, while the negative values indicated the inverse association with urban sprawl. However, a positive association of the variables indicated the probability of a decrease in the urban sprawl, and a negative association of the variables indicated the probability of increasing the urban sprawl.

In this case, 13 variables showed a positive association and 7 variables showed a negative association in the analysis. Policy regulations and regional accessibility had a strong positive relationship of 0.90 and 0.72, respectively, with the probability of urban sprawl. Policies in the city seem to be loose, which leads to illegal housing development, illegal land subdivision, and illegal land ownership in the city. The maximum extent of living space is not defined in the policy guidelines that lead to low-density sprawling development. In addition, the lack of rigor in taxation causes the illegal occupancy of land in the municipality, which further increases sprawling growth. Meanwhile, the regional availability in the city is relatively high compared to the other areas in the Batticaloa district. The cost of accessibility in rural areas is relatively high due to inadequate infrastructure facilities that cause migration to the city. People chose affordable areas of the city where home values were low. For example, the Thiraimadu, Saththurukondan, and Thiruperunthurai areas have a quite low land value compared to other areas of the city. Furthermore, the poor accessibility in the current land use patterns in the Batticaloa district causes people to settle in the city or urban areas. This inefficient land use pattern leads to more increased urban sprawl in the municipality.

Furthermore, the distance to a highway, distance to a secondary road, distance to a railway, and distance to a commercial area have a very weak effect on urban sprawl. The roads in the municipality were well laid out in most areas of the city. The municipality's transportation network is comparatively accessible to all, which can be one of the reasons for the very weak relationship with urban sprawl. Moreover, land value, demographic dynamics, and population by GN division have a strong negative relationship, with −0.7519, −0.5519, and −0.5766 coefficient values, respectively. It means that these factors have a very

strong effect on urban sprawl. The land value is not reasonable for all classes of people to buy land in the city. Rising house or land prices in core cities such as Puliyanthivu, Arasady, Kottamunai, and Thamaraikeny are forcing people to buy land in affordable areas such as Saththurukondan, Thiraimadu, Paalameenmadu, Kokkuvil, and Thiruperunthurai. Thus, these areas are developed as low-density, dispersed, and discontinuous sprawl in the municipality. Due to insufficient housing programs, people have owned the individual land in the city, which is increasing land prices by multiple magnitudes. Simultaneously, increasing demand for land is driving future real investments in property in the city, and land value expectations among owners are delaying land sales until a satisfactory price is obtained. Hence, the vacant spaces in the city remain without sales. Furthermore, demographic changes are significantly associated with urban sprawl. Migration from the rural areas to the city has increased the dynamics of population change in recent decades. Migrants to the city during the civil war tend to live in urban housing. It forms a sprawling growth pattern in the city under the demands of social status. At the same time, the standard of living of people from the districts encourages them to migrate to the Batticaloa municipality. Additionally, proximity to CBD, distance to educational resources, and distance to religious resources have a very weak effect on urban sprawl. These areas are mainly located in the core city, which is the reason for the very weak effect on urban sprawl. In contrast, the model does not show a significant relationship between the built-up areas and some socioeconomic factors such as housing preferences, income inequality, education, living standards, employment, and distance to air runways in the Batticaloa municipality. Simultaneously, these six factors did not meet the requirement of Cramer's V of 0.15. However, these factors may influence urban sprawl in the future depending on people's interferences and urban development. However, this study integrated the significant factors only in the prediction of urban sprawl.

### 3.3. Transition Potentials

A probability map for a transition area was created using logistic regression to predict the pattern of urban sprawl in 2030. This transition map shows a visual difference in suitability or specific potential for change over time for the built-up area (see Figure 7). The map illustrates that built-up areas tend to happen in the outskirts of the municipality. Simultaneously, the occurrence of built-up areas has the lowest probability in the city center, such as in Puliyanthivu and Arasady.

### 3.4. Urban Sprawl Prediction in 2030

The built-up areas are the primary parameters to understand urban sprawl. This built-up class includes residential buildings, commercial and shopping centers, highways and major streets, industrial areas, and other related properties [59]. The urban sprawl pattern for 2030 was simulated based on the built-up images (1990–2020) and several explanatory factors that included biophysical and socioeconomic factors. The built-up pattern was predicted in three scenarios. Scenario 1 indicated the built-up prediction for 2030 only with biophysical factors. Scenario 2 specified the built-up prediction for 2030 only with socioeconomic factors. Scenario 3 showed the built-up prediction for 2030 with biophysical and socioeconomic factors (see Figure 6). Furthermore, the probability of changes in the built-up areas in 2030 is presented in Table 5. The built-up area will probably change to non-built-up at around 7.97%, while the non-built-up area to built-up area can be 3.97% in 2030.

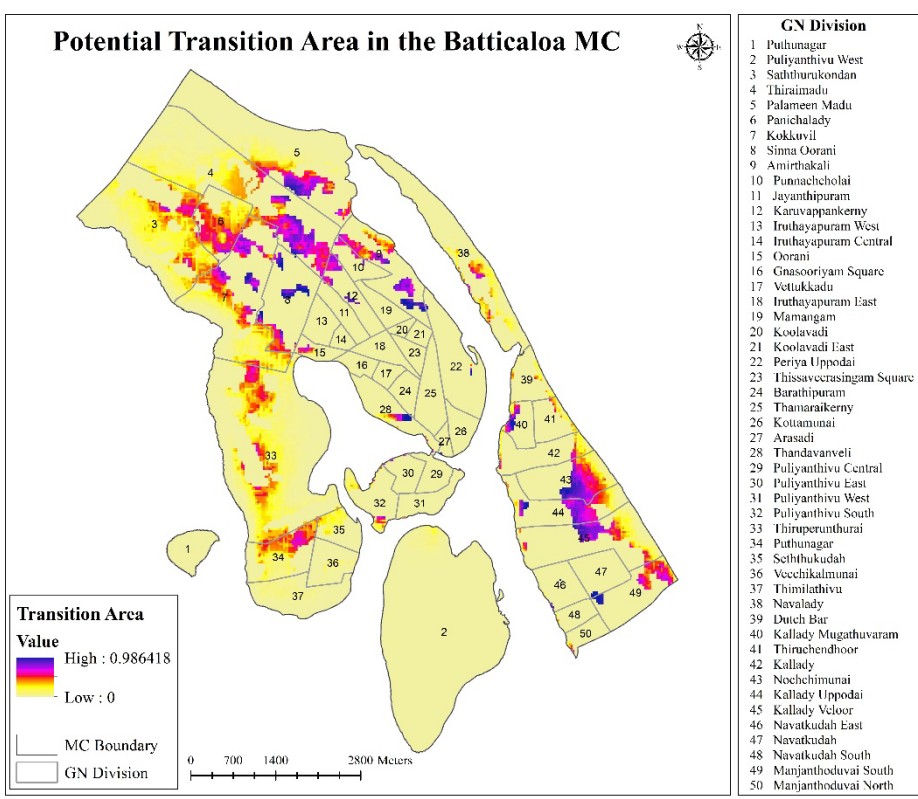

**Figure 7.** Potential transition area in the Batticaloa municipality.

**Table 5.** Probability of changes in the built-up area in 2030.

|  | **Built-Up** | **Non-Built-Up** |
|---|---|---|
| Built-up | 0.9203 | 0.0797 |
| Non-built-up | 0.0397 | 0.9603 |

The level of kappa agreement was assessed for the simulated and existing built-up maps in 2020 using the validate tool in IDRISI 17.0. The kappa statistics for K_no (0.8550), K_standard (0.8459), and K_location (0.8572) were identified above 0.80, indicating an excellent agreement between the datasets [49]. Thus, the transition probability matrix can be considered to apply in the prediction of built-up patterns in 2030. Furthermore, the ROC statistic was 0.9683, which designated a very strong value, and the soft prediction was precise. The pseudo R2 value was 0.6061, which specified a relatively good fit of the model because a pseudo R2 value greater than 0.2 is considered a relatively good fit.

Then, the prediction of urban sprawl was simulated in three scenarios for Batticaloa municipality. The result revealed that these three scenarios will have different extents of the built-up area in 2030. Scenario 1 indicated that the built-up area can be 1904.28 hectares in 2030. However, scenario 2 assumed that the built-up area could be 2118.24 hectares, which is relatively higher than scenario 1. In addition, the built-up area in scenario 3 showed approximately 2133.63 hectares, which was more than in scenarios 1 and 2 (see Table 6 and Figure 8). Scenario 1 was simulated with only biophysical factors (elevation, slope, and distance to water bodies), and scenario 2 was simulated only with socioeconomic factors (land value, demographic dynamics, policy regulations, regional accessibility, population by GN division, land value by GN division, distance to highway, distance to secondary road, distance to tertiary road, distance to railway, services buildings, proximity to CBD, distance to commercial, distance to educational, proximity to administration, distance to religious, proximity to services buildings, built-up cubic trend). This comparison confirmed that socioeconomic factors have a more significant influence on urban sprawl than biophysical

factors in the municipality based on the scenario-based findings. Thus, it can be concluded that urban sprawl is a socioeconomic phenomenon, which has a superior reflection on the built-up development of the city. Additionally, biophysical factors are also crucial in predicting urban sprawl, which has a significant effect on built-up growth. Therefore, the prediction of urban sprawl with biophysical and socioeconomic factors is more appropriate to display for future urban planning and development.

**Table 6.** Scenario based on built-up land area in 2030 (in hectares).

| Class | Scenario 1 | Scenario 2 | Scenario 3 |
|---|---|---|---|
| Built-up | 1904.28 | 2118.24 | 2133.63 |
| Non-built-up | 2290.90 | 2076.94 | 2061.55 |
| **Total** | **4195.18** | **4195.18** | **4195.18** |

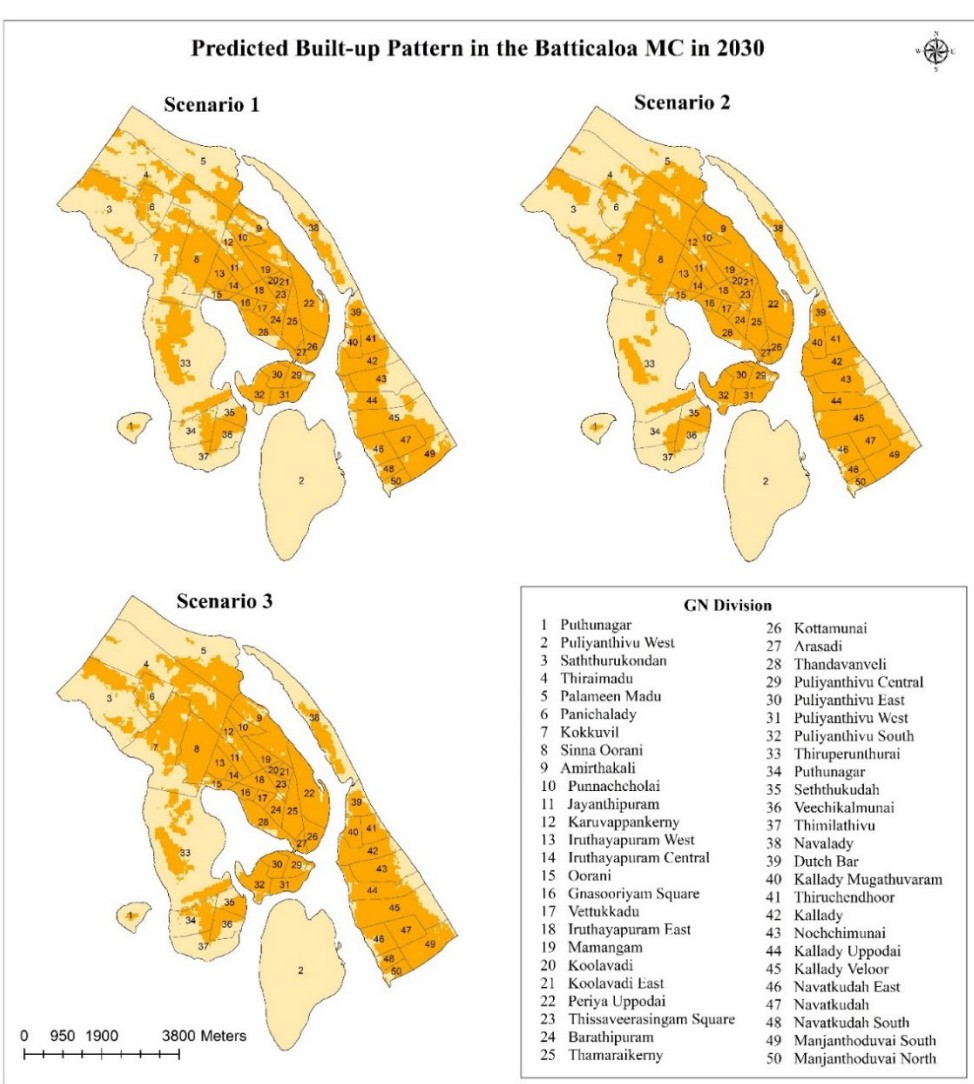

**Figure 8.** Scenarios based on predicted built-up patterns for urban sprawl in 2030.

In addition, the expected built-up growth in these three scenarios was identified between current (2020) and future (2030) patterns that showed different growth rates. When the built-up pattern is the same as in scenario 1 in 2030, the growth rate can be 0.73%; while the built-up pattern will be the same as in scenario 2 in 2030, the built-up growth rate can be 1.94%. The built-up pattern is the same as in scenario 3 in 2030, and the growth rate could be 2.02%. The sprawling growth moved in the direction of the northern and

western parts of the municipality. However, the lagoon in the western part of the city limits the sprawling growth. Overall, scenario 3 indicated high growth among these predicted built-up patterns. Based on the historical pattern of built-up areas (1990, 2000, 2010, 2020), it confirmed that the built-up pattern in 2030 is also relatively the same as in recent years. Therefore, when the potential factors can be similar to the current ones during 2030, urban sprawl can be expected to continue and be one of the predicted scenarios. However, the factors are supposed to be limited by the influence of institutional and political measures on built-up development, such as land management, land use planning, and land regulation. In the meantime, unpredictable drivers may sometimes affect the accuracy and the patterns of the model in the future.

The findings indicate that urban sprawl in the future is typically expected in the marginal areas in the city. Urban sprawl occurs due to the undeveloped, abandoned, and agricultural land in the municipality. This finding is similar to that of Grigorescu et al. [30] in Romania. This undeveloped land formed a leapfrog and scattered development with fewer houses and a small population. Ahrens and Lyons [8] stated that income growth, accessibility, and population were some of the significant determinants of urban sprawl in Ireland. However, income was not identified as a significant factor of urban sprawl in Batticaloa municipality. Furthermore, Batticaloa municipality has been experiencing a residential-based sprawling in core and marginal areas, which was similarly found in Hangzhou, China [60]. The demand of society and the socioeconomic conditions related to residential development are reasons for the urban growth and sprawling development in the city. Increasing land consumption is one of the causes of the urban sprawl growth in the municipality. Simultaneously, inadequate housing programs and increasing land value induce the people to settle in the cheaper land areas such as Kokkuvil, Sathurukondan, and Thiruperunthurai.

Furthermore, urban sprawl generally occurs due to the rapid urbanization or de-urbanization in the cities. Based on the Sri Lankan context, the largest cities are in Western Province, such as Colombo and Sri Jayawardenepura Kotte. At the same time, Kandy, Galle, and Kurunegala are some of the major cities in Sri Lanka. According to Amarawickrama [61], nearly 25% of Sri Lankans live in cities, which is expected to rise to nearly 65% by 2030; as a result of this, additional metropolitan areas will be required to accommodate the anticipated population growth. Thus, the Eastern Metro Region has chosen Batticaloa and Ampara as metro areas with 1 million people or more [62]. According to recent trends, most small- and medium-sized cities are growing fast as multi-functional centers. Several cities in Sri Lanka have strategies and plans for development, which pay little attention to the impacts of urban sprawl when implementing development projects [63].

However, the most pressing challenge confronting cities is urban sprawl [64], which has not been a focus of many urban studies in Sri Lanka [42,43,61,63]. According to the government's plans, Western Province is to be developed into an urban center in South Asia, such as Megapolis. This province in Sri Lanka has demonstrated well the extent of urban sprawl in the country [61,64]. The population in nine provincial capital cities grew by 6.42% annually between 1995 and 2017. It totaled roughly 7.39 million people in 2017, according to an analysis of satellite images undertaken for the State of Sri Lankan Cities' project in 2018. Sri Lanka's urbanization statistics show that the country has been de-urbanizing over the past 50 years. In 1987, after removing the unit of Town Councils, the country modified its municipal boundaries, resulting in an immediate drop in the population of the cities. Since then, the administratively designated urban population has grown modestly, with no changes to the city limits. However, areas that exhibit metropolitan spatial characteristics have grown rapidly, particularly on the outskirts of big cities [65]. Therefore, urbanization and de-urbanization are considerably influenced by urban sprawl growth.

## 4. Conclusions

This study attempted to predict the urban sprawl pattern in 2030 by integrating the socioeconomic and biophysical factors in the Batticaloa municipality. The constructed built-up area was obtained for the years 1990, 2000, 2010, and 2020 using NDBI analysis

in ArcGIS 10.6.1. This pattern confirmed that Batticaloa municipality has experienced the growth of urban sprawl since 1990 (see Figure 3). The built-up area was increased to 906 hectares between 1990 and 2000, which represents a large increase in the built-up growth in the municipality.

In addition, a total of 20 potential variables for socioeconomic and biophysical factors were identified from Cramer's V test. These variables were included to produce the transition probability map for prediction purposes. Of these, a socioeconomic factor is the most attractive driver, such as policy regulation, which has shown a high positive correlation. The land value has an inverse association among the variables. It means that it has a high probability of urban sprawl in the municipality. In addition, the urban sprawl spread to all parts of the municipality without being concentrated in a particular area. Several highly ranked factors for urban sprawl affect the entire municipality, such as land value, demographic dynamics, policy regulations, and regional accessibility.

The method used can integrate several factors for urban sprawl, which was an advantage in this study. Three prediction scenarios for urban sprawl were obtained from the past built-up maps and the transition map. Higher growth is expected under scenario 3, compared with scenarios 1 and 2 because scenario 3 was simulated with biophysical and socioeconomic factors. In the comparison between scenarios 1 and 2, scenario 2 showed a higher built-up growth that was simulated only with socioeconomic factors. However, the finding is that biophysical and socioeconomic factors are the most significant parameters in the growth of urban sprawl in Batticaloa municipality.

Furthermore, the spatial model was applied with a number of explanatory factors that can provide essential datasets on the measurement, quantification, and detection of urban sprawl. Additionally, the spatial results can be favorable for transport planning and planning-related services. Meanwhile, identifying the drivers of urban sprawl is crucial for understanding the sprawl in urban areas. Hence, in all the aspects mentioned above, it can aid in addressing urban sprawl on different spatial and temporal scales and help urban planners and decision makers improve development strategies in the municipality. The municipality can aim to reduce land abandonment and undeveloped areas, which are some of the main features of urban sprawl. Furthermore, predicted maps with different scenarios can support evaluating future sprawling growth and use to develop sustainable planning for the city. It can also provide a direction for future studies on urban sprawl modeling. In addition, this study used a hybrid model to integrate socioeconomic factors into the model for prediction. This technique can assist scholars in how to handle socioeconomic factors of the urban sprawl in the future. Moreover, the method used in this study is appropriate to predict the urban sprawl in any city in developed and developing countries.

In addition, as Batticaloa is one of the medium-sized cities in Sri Lanka, all the selected socioeconomic factors were not significant with the built-up pattern. Even though the selected socioeconomic factors were identified in different cities in developed and developing countries in the world, it should be applied in other cities in Sri Lanka, which can give more meaningful results for future development of cities. In addition, this study used the factors applied in a medium-sized city (Batticaloa) in Sri Lanka, which should be focused on large cities in Sri Lanka and the world. The prediction of urban sprawl should consider many socioeconomic factors according to the urban context, which can be useful to minimize the effects in the future. Finally, the overall findings could contribute to the sustainable development goal 11.3.1, which means that it can help to improve sustainable urban development by creating the proper guidelines to control urban sprawl in a city. Public engagement on urban sprawl and its driving factors can provide an understanding of how to overcome this problem in the future. It can also provide knowledge about sustainable housing developments to the public.

**Author Contributions:** Conceptualization, methodology, Mathanraj Seevarethnam and Noradila Rusli; software, Mathanraj Seevarethnam; validation, Mathanraj Seevarethnam, Noradila Rusli, and Gabriel Hoh Teck Ling; formal analysis, Mathanraj Seevarethnam; writing—original draft preparation, Mathanraj Seevarethnam; writing—review and editing, Mathanraj Seevarethnam, Noradila Rusli, and

Gabriel Hoh Teck Ling; supervision, Noradila Rusli and Gabriel Hoh Teck Ling; funding acquisition, Noradila Rusli. All authors have read and agreed to the published version of the manuscript.

**Funding:** This research received no external funding.

**Institutional Review Board Statement:** Not applicable.

**Informed Consent Statement:** Not applicable.

**Data Availability Statement:** Not applicable.

**Acknowledgments:** Our gratitude goes to Ismail Said, who has provided constructive comments and suggestions on this manuscript.

**Conflicts of Interest:** The authors declare no conflict of interest.

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
