# Peer review of "Prediction of Urban Sprawl by Integrating Socioeconomic Factors in the Batticaloa Municipal Council, Sri Lanka"

_ijgi, doi:10.3390/ijgi11080442_

Round 1

Reviewer 1 Report

Dear Author(s), I would like to thank you for the opportunity to read your manuscript entitled “Prediction of Urban Sprawl by Integrating Socioeconomic Factors in the Batticaloa Municipal Council, Sri Lanka”.

The overall manuscript is well presented with minor spelling or grammar mistakes.

The overall work is very interesting, as the sprawl model that allows for future prediction of urban sprawl patterns is very relevant and necessary especially in developing cities like Batticaloa in Sri Lanka.  

Here are some issues concerning your paper:

1.      The overall purpose of the article is stated clearly in the Introduction (lines 117-118) and should also be underlined in the Abstract.

2.      The literature review part presented in Introduction is logical and well organized. The review part based on the analysis of urban models and conditions is good. This part needs no improvement as the gap knowledge is well explained.

3.      The Materials and Methods part is clear. The method and tools are well-chosen. It is worth underlining the scientific soundness as very high due to multiple (NDBI, Cramer’s V, logistic regression, and CA-Markov analysis) methods used in the research.

4.      All maps in Figures are well presented and readable.

5.      According to the chart in Figure 5: what is on the unit of Gains and Losses on OX axis?

6.      I see misunderstanding about socioeconomic and biophysical factors identified for Your research. In the abstract You write there were 20 of them (line 19 and line 550), further in the text You write approximately 26 (line 356) and in the Figure 6: 25 of them are shown. Please determine their number. What is more important please justify why these specific were chosen. “factors related to urban sprawl were identified from the literature that was used to develop the transition potential for the urban sprawl prediction” (lines 356-358) seems insufficient, as it is not known whether these are universal factors and whether they reflect the analysed area of Batticaloa well.

7.      The Results part is logical and supported by literature base.

8.      As conclusions You write: “This comparison confirmed that socioeconomic factors have a more significant influence on urban sprawl than biophysical factors in the municipality.” (lines) In the meantime, the “aim to predict the urban sprawl pattern in 2030 by integrating socioeconomic and biophysical factors” (lines 117-118). Besides, Scenario 1 was simulated with biophysical factors, while Scenario 2 with socioeconomic factors. The factors do not seem to be integrated in the prediction – reconsider the goal again – see point 1 of this review.

9.      Future research directions and the significance of the results of the research achieved should be underlined and explained in conclusion part.

Reviewer

Author Response

Dear Reviewer,

Thank you very much for your comments and suggestions. We have revised the article based on the comments.

Thank you.

Authors.

Reviewer 2 Report

The presented article is relevant for Sri Lanka, which is facing the problem of urbanization in special conditions of limited territory and resources. The article meets the requirements for scientific papers and can certainly be published. As comments that can be used by the authors to improve the article, I want to note the following: 1. Improve the readability of Figure 6; 2. Pay attention to the need for urbanization / deurbanization of the territory of Sri Lanka as a whole, 3. To indicate whether the proposed method can be used for other countries or is it adapted only for the purposes of the country?

Author Response

(The authors gave the same response as above.)

Reviewer 3 Report

Review on the paper entitled “Prediction of Urban Sprawl by Integrating Socioeconomic Factors in the Batticaloa Municipal Council, Sri Lanka”.

In this paper, the authors attempt to predict the urban sprawl pattern in 2030 by integrating the socioeconomic and biophysical factors in the Batticaloa municipality, Sri Lanka. The paper is basically well-structured and -written, the figures are of high quality and the tables are necessary components of the paper. Whereas the study focuses on a local issue, some parts of it (e.g., the methodology) can be draw the attention of international readership. I think the paper is scientifically sound. However, I recommend the authors to revise the paper and improve its quality. The list above contains my observations regarding the paper:

1) In some cases, the authors refer to previous or other studies without citing them, for example, in lines 52 (“Limited studies were conducted…”) and 106 (“In the Sri Lankan context, urban sprawl studies are…”). Please, include references in the text.

2) In many cases, the authors put the numbers in parentheses after writing them in words like here (line 124): “…civil war for three (3) decades…”. This happens in lines 114, 187, 261, 303, 313, 369, 370, 448, 550, and 560. I think, under ten, the numbers should be written in words, and above ten in figures.

3) In line 154, the authors refer to a questionnaire survey without attaching the questionnaire to the paper. I recommend them to upload the questionnaire as a supplementary material.

4) In lines 123-124, the authors write that “The total population is almost 94,131…” I think the authors indicated the exact population data rather than what can be described with the term “almost”.

5) Also here (lines 124-125), this can be read: “The total extent of Batticaloa municipality is approximately 4,195.18 hectares…” This is a quite precise data rather than an approximate one.

6) Table 2 is not clear. Please add more information about the table. In addition, it is not clear who scaled the socioeconomic factors: the authors or the respondents?

7) In line 170, this can be read: “…the following equation 1…”. What does “1” mean here?

8) In lines 191-193, the authors write that “These 400 questionnaires were collected with the geographical coordinates of the location where the respondent filled in the form to understand the level of influence of each socioeconomic factor.” Does it mean the all the respondents filled out the form in their dwellings?

9) The authors assert (line 309) that “…the predicted map for 2030 can be considered to be accurate”. I agree with them, if we accept that there will be no other socioeconomic factors in the future affecting urban sprawl. For example, as far as I know, Sri Lanka has recently run out of gasoline making transportation and commuting difficult and thus increasing the significance of such factors as proximity and accessibility in the city. What I want to say here is that due to the increasing number of unpredictable variables (e.g., pandemics, wars, energy crisis), the prediction of the future is becoming more and more difficult.  

10) In lines 330-331, this can be read: “…Batticaloa experienced the effects of civil war in the 1980s and 2000s…”. It is not clear what those effects were and how they affected the migration in Batticaloa.

11) In line 356, the authors write that “Approximately 26 biophysical and socioeconomic factors related to urban sprawl were identified…”. Actually, they exactly identified 26 such factors. In addition, in Figure 6, only 25 factors are visualized.   

12) In lines 445-448, the authors add that “…the model does not show a significant relationship between the built-up and some socioeconomic factors such as housing preferences, income inequality, education, living standards, employment and distance to air-runway in the Batticaloa municipality.” It must be noted that these factors might seem to be less important at the moment, but their significance can change any time in the future (see my ninth comment). 

Author Response

(The authors gave the same response as above.)

Round 2

Reviewer 3 Report

Review on the revised version of the paper entitled “Prediction of Urban Sprawl by Integrating Socioeconomic Factors in the Batticaloa Municipal Council, Sri Lanka”.

The authors have carefully addressed all my comments I raised in my previous review report. I think the paper is now suitable for publication in the journal.

However, I recommend that the authors should modify or remove the text in the “Acknowledgements” section located at the end of the paper because it currently sounds this way: “In this section, you can acknowledge any support given which is not covered by the author contribution or funding sections. This may include administrative and technical support, or donations in kind (e.g., materials used for experiments).” Please check it.